# Using Genetics to Assess the Role of Acetate in Ischemic Heart Disease, Diabetes, and Sex-Hormone-Related Cancers: A Mendelian Randomization Study

**DOI:** 10.3390/nu16213674

**Published:** 2024-10-29

**Authors:** Jie V. Zhao, Junmeng Zhang

**Affiliations:** 1School of Public Health, Li Ka Shing Faculty of Medicine, The University of Hong Kong, Hong Kong SAR, China; junmengz@connect.hku.hk; 2State Key Laboratory of Pharmaceutical Biotechnology, The University of Hong Kong, Hong Kong SAR, China

**Keywords:** acetate, heart disease, cancer, short-chain fatty acid

## Abstract

Background: Acetate, a short-chain fatty acid, has gained attention for its contrasting roles, with evidence suggesting it may offer cardiovascular protection but also promote cancer, particularly those involving sex hormones. However, these influences have been scarcely assessed in epidemiological research. Objective: To investigate the relationship between acetate and ischemic heart disease (IHD), diabetes, and cancers related to sex hormones. Methods: Mendelian randomization (MR) was used to assess potential causal effects, selecting genetic variants without linkage disequilibrium (r^2^ < 0.001) and with genome-wide significance for acetate (*p* < 5 × 10^−8^). These variants were applied to large genome-wide association studies (GWAS) for ischemic heart disease (IHD; up to 154,373 cases), diabetes (109,731 cases), and five sex-hormone-related cancers (breast, colorectal, prostate, ovarian, and endometrial cancers, ranging from 8679 to 122,977 cases). We employed various methods for analysis, including penalized inverse variance weighting (pIVW), inverse variance weighting, weighted mode, and weighted median. Results: This study indicates that acetate may be associated with a lower risk of ischemic heart disease (IHD), with an odds ratio (OR) of 0.62 per standard deviation (SD) increase in acetate and a 95% confidence interval (CI) of 0.39 to 0.98. Additionally, acetate was linked to a higher breast cancer risk, with an OR of 1.26 and a 95% CI ranging from 1.08 to 1.46. This association remained robust across multiple sensitivity analyses. Conclusions: Acetate, along with factors that influence its activity, may serve as possible targets for breast cancer treatment and possibly IHD, offering opportunities for new drug development.

## 1. Introduction

Ischemic heart disease (IHD) ranks as the leading cause of death worldwide [1]. Diabetes and cancer often co-occur with IHD [2] and also impose a heavy burden on public health. As such, it is in the public interest to find more modifiable targets for the prevention and intervention of IHD. Acetate is one of the modifiable targets. In recent years, acetate, a prominent short-chain fatty acid (SCFA), has gained growing attention due to its dual role [3]. On one hand, it has been regarded as a beneficial dietary component produced by the fermentation of dietary fiber [4], with possible cardiovascular health advantages. On the other hand, acetate plays a crucial role in energy production, promoting biomass growth in fast-dividing cancer cells, which raises concerns about its potential to increase the risk of cancer, particularly those related to sex hormones. Given that these effects have seldom been studied in humans, the primary goal of this research was to explore the overall and sex-specific associations between acetate and ischemic heart disease (IHD), diabetes, and cancers related to sex hormones.

Acetate can be obtained exogenously, for example, derived from the fermentation of fermentable dietary fiber, such as inulin. Acetate can also be generated endogenously, from the deacetylation of proteins [5]. A growing body of research indicates that fermentable dietary fiber may influence the gut microbiome [6] and provide positive cardiometabolic benefits [7,8,9,10]. For example, inulin increases the *Bifidobacteria* populations in adults [6] and lowers LDL-cholesterol, triglycerides, and fasting glucose [10]. Consistently, acetate supplementation improves lipid profile [4] and cardiac function in animal experiments [11]. A systematic review and meta-analysis in humans suggested that acetate supplementation could help reduce triglycerides and fasting-glucose levels among people with type 2 diabetes in the short term, although further validation of these findings is necessary [12]. Moreover, acetate may regulate sex hormones. An animal experiment has shown that acetate can attenuate androgen excess and increase sex hormone binding globulin (SHBG) in polycystic ovary syndrome (PCOS) [13,14]. As shown in previous Mendelian randomization (MR) studies [15,16], androgens and SHBG play an important role in IHD and diabetes, with sex disparity. So, we hypothesized that acetate might affect IHD and diabetes in a sex-specific way.

Acetate is considered to have a role in cancer, participating in both energy derivation and protein acetylation throughout all subcellular compartments [3]. In vivo experiments suggest that acetate promotes the survival of cancer cell [17]; however, the effect has not been examined in epidemiological studies. Considering the complexity of various cancer types and acetate’s possible influence on sex hormones, we primarily concentrated on its effects on sex-hormone-related cancers, such as prostate [16], breast [18], endometrial [19], ovarian [20], and colorectal cancers [21].

In the context of no randomized controlled trial available and the ethical concerns due to the potential harm, Mendelian randomization (MR) provides a method for investigating causal relationships without requiring direct intervention. As genetics are distributed randomly, not dependent on social factors or health status, this study design can be used to provide unconfounded association [22]. Although there are a few MR studies on metabolites and cardiometabolic diseases [23,24] or cancer [25], they used different metabolic panels which did not include acetate [23,24], and/or did not focus on ischemic heart disease (IHD) and cancer [23]. In this study, we used MR, for the first time, to test its role in IHD, diabetes, and several cancers related to sex hormones, including prostate, breast, endometrial, ovarian, and colorectal cancers. To be comprehensive, we also examined its sex-specific links with IHD and diabetes, along with its impact on their shared risk factors, including body mass index (BMI), lipids, blood pressure, glucose, and HbA_1c_.

## 2. Materials and Methods

### 2.1. Study Design

A two-sample MR design was employed in this study, and the datasets used are listed in Table 1. In particular, we employed genetic proxies for acetate from large GWAS related to IHD, diabetes, sex-hormone-related cancers, and their risk factors (lipids, glucose, HbA_1c_, blood pressure, and BMI). To investigate potential differences by sex using sex-specific analyses, we derived the sex-stratified associations with acetate from individual data in the UK Biobank and sex-specific outcome associations from the UK Biobank or large consortia. The study design was illustrated in the flow chart (Appendix A).

### 2.2. Genetic Instruments for Acetate

We chose single-nucleotide polymorphisms (SNPs) that show strong (*p* value < 5 × 10^−8^) and independent (linkage disequilibrium threshold of 1 × 10^−3^) associations with acetate as instruments. We obtained these SNPs from the UK Biobank. Summary statistics on the GWAS was provided by the IEU OpenGWAS project [36]. Metabolomics, including acetate, was assessed in 121,577 randomly selected samples with quality control. In the GWAS, all the samples used in the study were drawn from participants of European descent and acetate standardization was applied before conducting the analyses. The validity of the chosen SNPs was verified by calculating the F-statistic using a widely recognized formula, i.e., the square of beta (SNP-acetate)/se (SNP-acetate) [37]. A threshold of 10 was applied as a general guideline to differentiate strong instruments from weak ones [38]. The detailed information of the selected SNPs was shown in Appendix A. We also evaluated the relationship between these selected genetic variants and potential confounding factors, such as socioeconomic status and unhealthy lifestyle, in Phenoscanner, a dataset of comprehensive genotype-phenotype associations in humans “http://www.phenoscanner.medschl.cam.ac.uk/ (accessed on 12 January 2024)”. We assessed the associations between various possible confounders, including the Townsend index, education, smoking habits, alcohol intake, and levels of physical activity, using GWAS summary data from the UK Biobank. The summary data were collected from Neale Lab (http://www.nealelab.is/uk-biobank). We removed genetic variants associated with any of these factors at genome-wide significance (*p* < 5 × 10^−8^) in sensitivity analyses. Rs1260326 was associated with alcohol drinking in both Phenoscanner and the UK Biobank (Appendix A), and rs3184504 was related to smoking status in Phenoscanner. As acetate is related to alcohol intake [5], it is also possible that the pleiotropic association is vertical pleiotropy; so, we did not remove it in the main analysis. When checking their associations with the outcomes, we found rs3184504 was also related to IHD, diabetes, and endometrial cancer, and rs1260326 was also related to diabetes. So, we performed colocalization analyses; if there was an indication that rs1260326 or rs3184504 were shared causal variants of acetate with these outcomes, we kept the SNP(s); otherwise, we removed them in the sensitivity analysis. Colocalization tested four hypotheses: H_0_: no association with either trait; H_1_: association with only trait 1; H_2_: association with only trait 2; H_3_: two independent SNPs associated with both traits; H_4_: the same SNP associated with both traits [39]. A high posterior probability for shared causal variants (PP_H4_) suggests colocalization, whereas a high PP_H3_ indicates confounding [39]. In the check using Phenoscanner, we found the two SNPs (rs1260326 and rs3184504) were also related to triglycerides and LDL-cholesterol respectively. In addition, we found rs28601761 was related to triglycerides. As triglycerides may be mediators from acetate to IHD, we did not remove rs28601761 in the main analysis.

Primary outcome: IHD, diabetes, and sex-hormone-related cancers, including prostate, breast (including overall, ER+, ER- breast cancer), endometrial, ovarian, and colorectal cancers.

Secondary outcomes: lipids (including triglycerides, LDL-cholesterol, apolipoprotein B), glucose, HbA_1c_, blood pressure, and BMI.

### 2.3. Genetic Associations with the Primary Outcomes in Overall and Sex-Specific Analyses

In the overall analysis, we derived a summary of genetic links with IHD from the CARDIoGRAMplusC4D consortium, which included 122,733 cases and 424,528 controls, primarily of European ancestry [26]. Additionally, genetic associations with IHD were obtained from the FinnGen study with 31,640 cases and 187,152 controls. To enhance statistical power, MR estimates from both GWAS studies were combined through meta-analysis. Genetic associations for type 2 diabetes (referred to as “diabetes”) were gathered from DIAGRAM, which comprised 74,124 cases and 824,006 controls [27], and FinnGen study of 35,607 cases and 183,185 controls. Similarly, MR estimates for diabetes were meta-analyzed using both GWAS datasets (Table 1).

We acquired genetic associations for prostate cancer from the PRACTICAL consortium (Prostate Cancer Association Group to Investigate Cancer-Associated Alterations in the Genome), which included 79,148 cases and 61,106 controls, adjusting for principal components and relevant study covariates [28]. For breast cancer, data were sourced from the Breast Cancer Association Consortium (BCAC) [29], adjusting for country and principal components. Genetic associations with endometrial cancer were obtained from a large GWAS meta-analysis, which included 12,906 cases and 108,979 controls, also adjusting for principal components [30]. For ovarian cancer, the data came from the Ovarian Cancer Association Consortium (OCAC), comprising 25,509 cases and 40,941 controls, again adjusted for principal components [31]. Lastly, genetic associations for colorectal cancer were sourced from FinnGen, which included 3022 cases and 215,770 controls.

In the sex-specific analysis of IHD, we employed individual-level data from the UK Biobank (application number 42468) [40]. A total of 502,713 participants, aged 40–69 years, mean age of 56.5 years, with 45.6% being men, were recruited between 2006 and 2010 in England, Scotland, and Wales, with the majority (94%) being of self-reported European ancestry. In the analysis, we restricted to the participants who are white and British. Genotyping was assessed using the UK BiLEVE array and UK Biobank Axiom array. For quality control, as previously described [41], we excluded participants who reported inconsistent information about sex, participants with an excessive level of relatedness (having more than 10 potential third-degree relatives), participants with abnormal sex chromosomes (such as XXY), participants with low-quality genotyping, determined by heterozygosity and missing data rates, as well as those who had withdrawn. IHD events were identified through linkage to hospitalization and death records, along with a nurse-led interview conducted at the time of recruitment. Following quality control procedures, 47,413 cases of IHD were identified, with 31,127 men and 16,286 women. Logistic regression was performed, adjusting for age, assay array, and 20 principal components, to determine sex-specific associations with IHD in both men and women. In diabetes, we obtained the sex-specific associations with diabetes from summary statistics of a GWAS meta-analysis, which included 30,053 cases and 434,336 controls in women, and 41,846 cases and 383,767 controls in men [27].

### 2.4. Genetic Associations with the Secondary Outcomes in Overall and Sex-Specific Analyses

We derived the genetic associations for lipids from the Global Lipids Genetics Consortium (GLGC), a large-scale consortium of over 0.84 million participants of European ancestry, excluding the UK Biobank [32]. Genetic associations with glucose and HbA_1c_ were sourced from the Meta-Analyses of Glucose and Insulin-related traits Consortium (MAGIC), including 140,595 individuals without diabetes for glucose [42] and up to 145,579 people for HbA_1c_ [34]. The associations with respect to blood pressure were provided by a large GWAS meta-analysis of the International Consortium of Blood Pressure (ICBP) and the UK Biobank [35].

We obtained sex-specific genetic associations for triglycerides and LDL-cholesterol from the Global Lipids Genetics Consortium (GLGC) [32]. Sex-stratified associations for BMI, blood pressure, glucose, and HbA_1c_ were derived from the UK Biobank GWAS summary data provided by Neale Lab, with adjustments for age, age squared, and 20 principal components.

### 2.5. Statistical Analysis

After removing palindromic SNPs and SNPs with mismatched alleles (shown in Appendix A), we calculated the Wald ratio of each individual SNP by dividing the genetic association with the outcomes by the association with the acetate. Afterward, we combined the Wald estimates by applying inverse variance weighting (IVW) using multiplicative random effects [43]. We presented MR estimates as odds ratio (OR) for primary outcomes (diabetes, IHD, and five sex-hormone-related cancers), as beta coefficient for other IHD risk factors per standard deviation increase in acetate. To test whether the detected associations were bi-directional, we also conducted bi-directional MR, in which we obtained genetic instruments for these outcomes, using a similar method as we used for acetate, and then examined the associations of genetically predicted outcomes with acetate. In addition to IVW, we also used penalized IVW (pIVW), a novel method that extends to conventional IVW by accounting for the weak instrument issue using the penalization approach and accounting for balanced horizontal pleiotropy but does not provide a test for directional pleiotropy [44]. Additionally, we used MR-Egger, MR-PRESSO, weighted median, and weighted mode, which also remain resilient to pleiotropy. The weighted median provides a robust estimate of causal effects, even if up to 50% of the genetic variants used as instruments are invalid [45]. The weighted mode assumes that most genetic variants serve as valid instruments, meaning no majority of invalid instruments estimate the same causal effect as the valid ones [46]. MR-Egger can assess whether genetic variants have pleiotropic effects on the outcome that differ on average from zero (directional pleiotropy) and provide a consistent estimate of the causal effect [47]. MR-PRESSO pinpointed the genetic variant(s) responsible for the differences in associations, referred to as outliers, and offered adjusted estimates by excluding these outliers [48]. To control for multiple testing, a Bonferroni-corrected cut-off, i.e., *p* value < 0.05/15 (7 primary outcomes + 8 secondary outcomes) = 0.003 was applied. Associations with a *p*-value < 0.05, but not reaching the Bonferroni-corrected *p* value, were considered as suggestive associations.

A power calculation was conducted. The sample size is determined by dividing the sample size for the exposure–outcome association by the r^2^ of the genetic proxies for the exposure [49,50]. In the situations where multiple GWAS of outcomes were used, power calculations were conducted using the combined sample size from the GWAS. The differences in estimates across various datasets were indicated by *I*^2^, calculated using a heterogeneity test in the “meta” package. The heterogeneity among the SNPs was reflected by Cochran’s Q statistic and heterogeneity *p* value, calculated using “MendelianRandomization” package. We tested the *p* values for sex differences with a heterogeneity test implemented in the “meta” package.

All statistical analyses were performed using the “TwoSampleMR”, “MendelianRandomization”, “mr.pIVW,” “coloc”, and “meta” packages in R (version 4.0.1, R Foundation for Statistical Computing, Vienna, Austria).

## 3. Results

### 3.1. The Role of Acetate in IHD, Diabetes, and Cancer

We identified nine SNPs for acetate, all of which had an F-statistic greater than 10, indicating they were strong instrumental variables (Appendix A). The genetic instruments selected for acetate explained 0.05% of the variance. Rs1260326 was linked to alcohol drinking in the UK Biobank (Appendix A), and rs3184504 was related to smoking status in Phenoscanner. Our main findings are that genetically predicted acetate is inversely related to IHD but positively associated with breast cancer (Figure 1). Our findings imply that genetically predicted acetate might be linked to a lower IHD risk (*p* = 0.04) but not with diabetes using IVW (Figure 2). The associations for IHD from the two data sources are consistent, with low heterogeneity (meta-analysis heterogeneity *I*^2^ = 0.0%, heterogeneity test *p* value = 0.996). The directions of association for diabetes were inconsistent; however, the confidence intervals from the two datasets overlap, and heterogeneity test did not show statistical difference (*I*^2^ = 42.2%, *p* value = 0.189).

IVW indicated that heterogeneity existed between the SNPs (heterogeneity test *p* < 0.05, Appendix A), implying that more robust methods against pleiotropy may yield more reliable estimates. The analysis using such methods, including pIVW, weighted mode weighted median, and MR-PRESSO, showed consistent results (Appendix A; MR-PRESSO detected outliers shown in Appendix A). Inconsistent with other methods, MR-Egger showed a quite wide confidence interval and null association (Appendix A) and did not suggest directional pleiotropy (Appendix A). The associations with IHD were suggestive rather than confirmative after Bonferroni correction; however, the effect size is smaller than the effect size detected by the current sample size, suggesting a sufficient power (Appendix A). Scatter plot and the leave-one-out analysis did not indicate the results were driven by a single SNP (Appendix A). The colocalization analysis showed that rs3184504 may be a shared variant with IHD (PPH3 = 0.013, PPH4 = 0.987 in Cardiogram, PPH3 = 0.006, PPH4 = 0.993 in FinnGen), and rs1260326 may be a shared variant with diabetes (PPH3 = 3.7 × 10^−4^, PPH4 = 1.00 in DIAGRAM, PPH3 = 0.043, PPH4 = 0.945 in FinnGen). So, in the sensitivity analysis for IHD, we did not remove rs3184504; in the sensitivity analysis for diabetes, we did not remove rs1260326. The association for IHD attenuated and included the null after excluding rs1260326 (Appendix A). Diabetes associations excluding rs3184504 were consistent with the main analysis (Appendix A). In the bi-directional analysis, genetically predicted IHD was related to acetate only in pIVW (*p* = 0.04) but not in other methods (Appendix A).

For the sex-specific results, we did not find apparent sex disparity (statistical test for sex difference has *p* > 0.05), and the confidence intervals overlap for men and women (Figure 3). Considering the potential pleiotropy, we also looked into sensitivity analyses using analytic methods that are more robust to pleiotropy and analysis excluding potentially pleiotropic SNPs.

Applying various analysis methods, the associations remained largely consistent (Appendix A), and after excluding rs1260326 for IHD and rs3184504 for diabetes (Appendix A), no outliers were detected in the scatter plot or leave-one-out plot (Appendix A).

Regarding cancer, genetically proxied acetate was associated with a higher risk of breast (*p* = 0.0025) and ER+ breast (*p* = 0.002), with a marginal relationship with ER- breast cancers (*p* = 0.05) (Figure 4).

The scatter plot and leave-one-out analyses did not indicate that the associations were driven by any single SNP (Appendix A). We cannot exclude a positive association with ER- breast cancer; as shown in Appendix A, the effect size may be below the minimum detectable effect size (at 80% power) by the current sample size. The associations with other cancers included the null. The associations showed consistent directions across different analytic methods (Appendix A). In the sensitivity analysis, we removed rs1260326 and rs3184504 for all cancers except endometrial cancer, where we only removed rs1260326, as rs3184504 showed high PP_H4_ (0.995) and low PP_H3_ (0.005) in the colocalization analysis with endometrial cancer. The associations were robust after removing pleiotropic SNP(s) (Appendix A), which provided some support for the observed associations. In the bi-directional analysis, genetically predicted breast cancer was not associated with acetate using different methods (Appendix A).

### 3.2. The Role of Acetate in Lipids, Glycemic Traits, Blood Pressure, and BMI

In the overall analysis, genetically proxied acetate was linked to lower ApoB using weighted median, higher HbA_1c_ using MR-PRESSO, higher SBP using weighted mode, lower DBP using MR-PRESSO, and lower BMI using weighted mode (Appendix A), but the associations were not shown in other methods, including IVW (Figure 5) and did not remain after Bonferroni correction. MR-PRESSO indicated some outliers (Appendix A), and the scatter plot also showed outliers for some associations, such as lipids, HbA_1c_, and BMI (Appendix A), but no single SNP appeared to drive the associations based on leave-one-out plot (Appendix A).

After removing rs1260326 and rs3184504, the associations were generally consistent, except that the associations with DBP were null using all methods (Appendix A). For the sex-specific result, we did not find sex disparity for these risk factors (Figure 6).

Genetically proxied acetate was related to reduced LDL-cholesterol levels using weighted median and weighted mode, lower triglycerides using weighed median, lower ApoB using weighted median and MR-Egger, higher fasting glucose and HbA_1c_ using MR-PRESSO, and lower BMI using weighted median and weighted mode in women (Appendix A), though none of these results remained significant after applying Bonferroni correction. It showed no associations with these risk factors, except for HbA_1c_ in men (Appendix A). Similarly, MR-PRESSO indicated some outliers (Appendix A), and the scatter plot also showed outliers for some associations, such as lipids, HbA_1c_, and BMI (Appendix A), but the leave-one-out plot showed that the associations were not driven by any single SNP (Appendix A).

## 4. Discussion

This study employs MR to assess the causal effect of acetate on IHD, diabetes, and sex-hormone-related cancers, providing key insights into underlying mechanisms and possible interventions. By minimizing confounding, MR findings indicate that acetate may lower IHD risk while increasing breast cancer risk, utilizing large-scale cohorts and consortia data. Consistent with animal experiments showing acetate supplementation improves lipid profile [4], our study suggests that acetate lowers triglycerides. The sex-specific analyses showed no sex disparity.

Our findings on breast cancer align with those of Wang et al. [25], who reported that acetate increased breast cancer risk amongst all metabolites. Nonetheless, their study was hypothesis-free rather than hypothesis-driven and did not examine the role of acetate in IHD and other cancers [25]. Our findings extend to previous evidence by providing support to the hypothesis that acetate may have dual roles in IHD and cancer. The beneficial cardiovascular effect of acetate was consistent with animal experiments showing that acetate supplementation improves cardiac function [11] and lipid profile [4]. Despite this, the association with lower risk of IHD needs to be interpreted more cautiously, given the inconsistent findings using different analytic methods and the attenuated association after excluding rs1260326. The association with lower triglycerides should be interpreted cautiously, since we only observed an inverse association in women using weighted median, and this relationship did not persist after multiple testing adjustment, indicating the possibility that the association with triglycerides occurs by chance.

Interestingly, we also found that higher genetically predicted acetate is linked to a higher breast cancer risk, including overall and ER+ breast cancer. We cannot rule out a possible relationship with an increased risk of ER- breast cancer, given the relatively smaller number of cases and limited statistical power. Several explanations might exist, and these findings should be interpreted with caution. It is possible that selection bias exists, i.e., people dying from breast cancer earlier before recruitment or dying from other diseases which occurred earlier from breast cancer, such as IHD, were not included in the GWAS of breast cancer, which may bias the estimates towards null or the other direction. However, using the GWAS of breast cancer in siblings in the UK Biobank, which is less susceptible to selection bias [51] but has less power due to the much smaller number of cases (16,586 cases) than the BCAC (122,977 cases), we obtained consistent directions of associations (using the weighted median method, an odds ratio of 1.31, *p* value of 0.10) as in the Breast Cancer Association Consortium. So, selection bias may not reverse the direction of association here. It is also possible that acetate affects inflammation, such as lowering NF-κB and TNF-α in animal experiments [13,52], which has a complex role in IHD and cancer. For instance, a recent MR finds that lower TNF-a is related to a reduced risk of IHD but an increased risk of breast cancer [53], mirroring the pattern observed in this MR study. The more obvious association between ER+ cancer versus ER- cancer might suggest a mechanism via estrogen. In animal experiments, acetate supplementation also increased 17β-estradiol in animals with PCOS [14]. Additional mechanistic studies are required to investigate the pathways involved.

By utilizing MR, this study investigates the hypothesis of the dual reputation of acetate, revealing new insights into its potential health effects. We utilized the largest available GWAS to date and applied MR to minimize confounding. We also acknowledge that several limitations exist. First, these findings were obtained from a European ancestry population, which may not be applicable to other ancestries like Asians and Africans. However, it is anticipated that the causal effects will remain stable across various settings. Second, as the genetic instruments capture only a limited fraction of the variation in exposure, i.e., acetate [49], the precision of MR estimates is lower than in conventional observational studies. However, we employed GWAS in large cohorts and consortia and, where possible, a meta-analysis of GWAS, to improve power. Despite the marginal association with ER- breast cancer, we cannot dismiss the possibility of an association with an effect size smaller than our current detection limits. Replication with a larger GWAS, when available, would be valuable. Third, the heterogeneity statistics (*I*^2^) must be approached with caution when the number of studies is limited [54]. However, the low *I*^2^ for IHD is consistent with the similar estimates from the two data sources (CARDIoGRAMplusC4D and FinnGen). Fourth, some associations showed large heterogeneity, possibly due to pleiotropy or different mechanisms by which genetic variants affect the exposure. In situations like these, estimates derived from methods that are more robust to potential pleiotropy are considered more reliable than those from IVW. Fifth, GWAS for fasting glucose and blood pressure have controlled for BMI, which may bias the estimates [55]. However, using the UK Biobank, which did not control for BMI, we found consistent, null associations (Appendix A). It would be ideal to have the same covariate adjustment in the GWAS of exposure and outcomes; however, it is difficult when using summary statistics from different cohorts and consortia, and there is no consensus about the covariates to control for. Sixth, sample overlap may introduce bias into the MR estimates [56]. In our study, the GWAS for acetate is from the UK Biobank. The GWAS for IHD in FinnGen and GWAS for colorectal cancer, prostate cancer, breast cancer, and ovarian cancer did not include UK Biobank, i.e., no sample overlapping. But in the GWAS for IHD from CARDIoGRAMplusC4D, the UK Biobank contributes approximately 28.1% of cases and 61.7% of controls. For the diabetes GWAS, the UK Biobank contributes approximately 25.8% of cases and 51.4% of controls. For endometrial cancer, the UK Biobank contributes 4.9% of cases and 57.7% of controls. Although we tried to minimize the sample overlapping, as UK Biobank is a valuable resource with large sample size, in some analyses, we cannot avoid the overlapping sample. However, a simulation study reports that sample overlap is not a significant issue when conducting two-sample Mendelian randomization analysis on large biobanks like the UK Biobank [57]. Moreover, MR analysis must follow certain strict assumptions: the genetic variants need to be related to the exposure, not linked to any confounders between the exposure and outcome, and must influence the outcome solely through the exposure, avoiding horizontal pleiotropy [22]. To ensure these assumptions were met, we chose SNPs that have a strong association with acetate. Population stratification could act as a possible confounder; however, all the GWAS used in this study were conducted with individuals of European ancestry, and genomic control measures were applied. Potentially pleiotropic SNPs related to lifestyle exist, such as rs1260326 associated with alcohol and rs3184504 related to smoking status in Phenoscanner. However, we cannot exclude the possibility of vertical pleiotropy for rs1260326, and the false positive association of rs3184504 with smoking, so we kept them in the main analyses and removed them in the sensitivity analyses. We also used colocalization to assist with the assumption testing and implemented multiple analytic methods that are robust against pleiotropy.

From a clinical and public health standpoint, our research suggests that acetate could be a promising target for breast cancer and IHD. Medications that boost acetate levels might offer protection against IHD and hyperlipidemia, but for women, this benefit should be balanced with the increased risk of breast cancer. Additionally, factors that regulate acetate or influence its pathways could be explored as possible targets for breast cancer therapies. For example, acetyl coenzyme A (CoA) synthetase 1 (ACSS1) and acetyl CoA synthetase 2 (ACSS2) have been shown to play a role in regulating how cancer cells utilize acetate [58], enhancing their ability to use acetate as an additional nutrition source [59]. ACSS2 is highly expressed in several cancers, including breast cancer, and ACSS2 inhibitors have been proposed to treat cancers in a patent by a drug discovery company, Metabomed Ltd., based in Yavne, Israel [60], with promising potential in the application to clinical practice.

## 5. Conclusions

To sum up, this MR research demonstrates that acetate is associated with a decreased IHD risk but an increased breast cancer risk. Future studies are needed to uncover the mechanisms underlying its dual reputation. Factors influencing acetate activity, like ACSS2, could be explored as potential drug targets for treating IHD and breast cancer.

## Figures and Tables

**Figure 1 nutrients-16-03674-f001:**
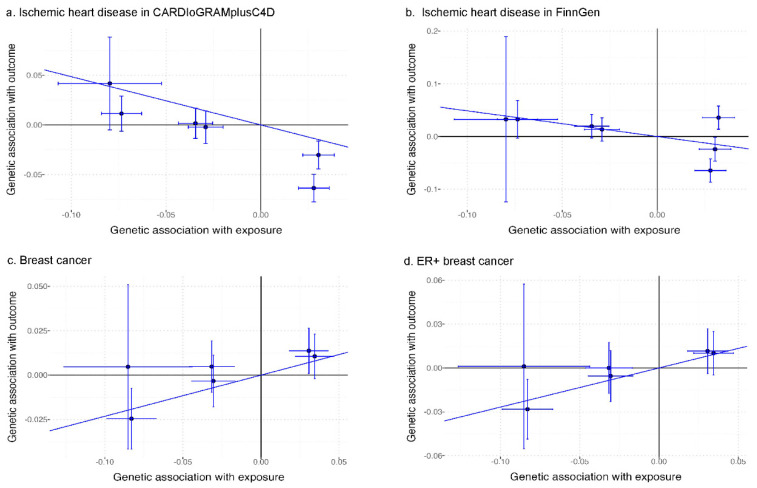
Associations of genetic instruments for acetate with ischemic heart disease and breast cancer.

**Figure 2 nutrients-16-03674-f002:**
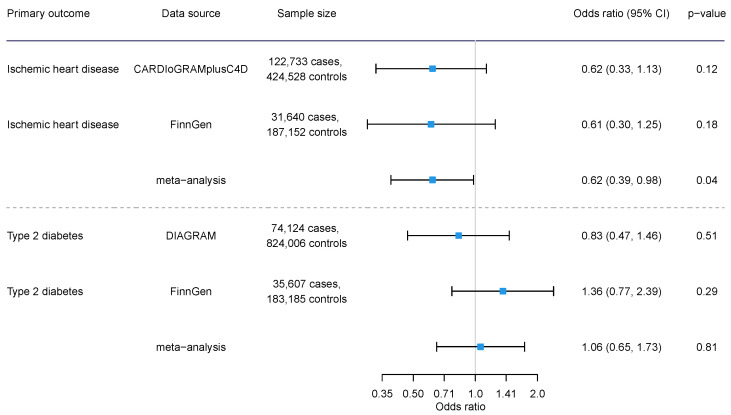
The overall association of genetically predicted acetate with ischemic heart disease and diabetes using inverse variance weighting.

**Figure 3 nutrients-16-03674-f003:**
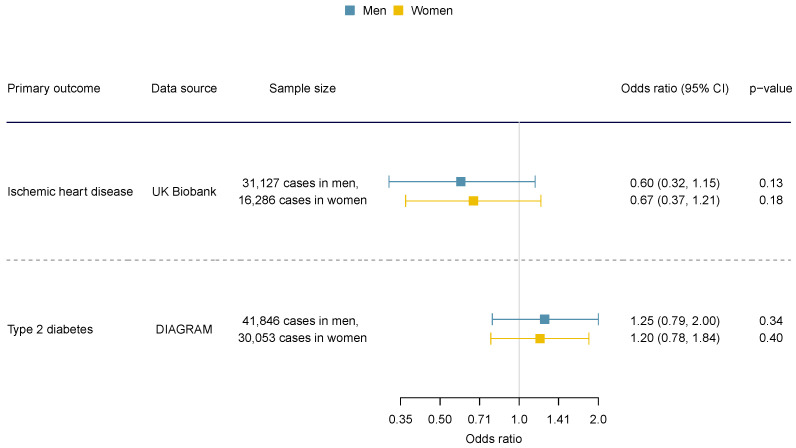
The sex-specific association of genetically predicted acetate with ischemic heart disease and diabetes.

**Figure 4 nutrients-16-03674-f004:**
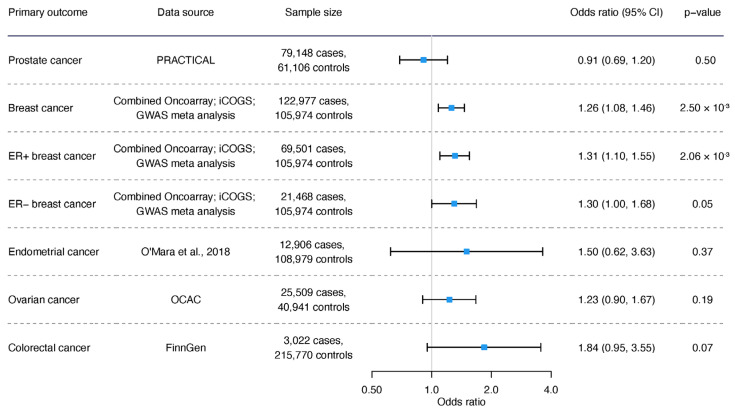
The association of genetically predicted acetate with prostate cancer, breast cancer (including overall and ER+, ER- breast cancer), endometrial cancer, ovarian cancer, and colorectal cancer [30].

**Figure 5 nutrients-16-03674-f005:**
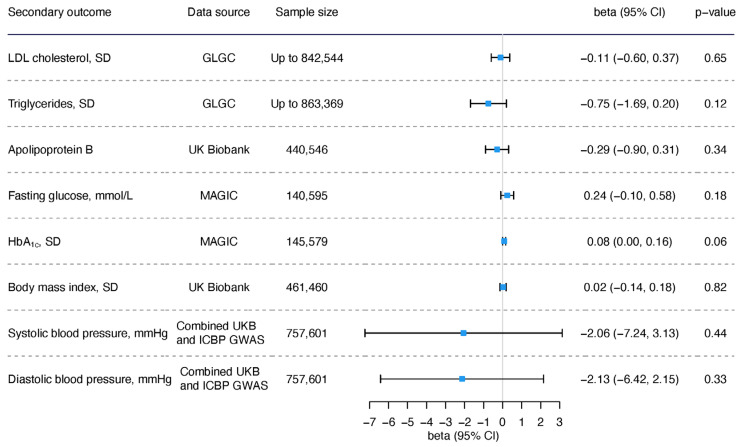
The overall association of genetically predicted acetate with lipids, glycemic traits, blood pressure, and body mass index.

**Figure 6 nutrients-16-03674-f006:**
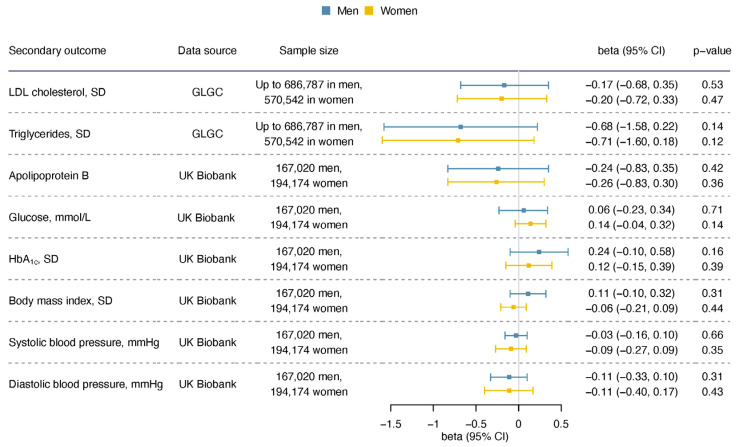
The sex-specific association of genetically predicted acetate with lipids, glycemic traits, blood pressure, and body mass index.

**Table 1 nutrients-16-03674-t001:** The datasets included in this study.

Outcomes	Genome-Wide Association Study (GWAS) Information			
	Sources	Sample Size	Covariates	Reference
Primary outcomes				
Overall				
IHD	CARDIoGRAMplusC4D	122,733 cases, 424,528 controls	Age, sex, the first 30 principal components, genotyping array	[26]
	FinnGen	31,640 cases, 187,152 controls	Age, sex, genotyping batch and the first 10 principal components	https://www.finngen.fi/en (accessed on 17 February 2024)
Type 2 diabetes	DIAGRAM	74,124 cases, 824,006 controls	Sex, study-specific covariates, population structure and relatedness	[27]
	FinnGen	35,607 cases, 183,185 controls	Age, sex, genotyping batch and the first 10 principal components	https://www.finngen.fi/en (accessed on 17 February 2024)
Colorectal cancer	FinnGen	3022 cases, 215,770 controls	Age, sex, genotyping batch and the first 10 principal components	https://www.finngen.fi/en (accessed on 17 February 2024)
Sex-specific				
IHD	UK Biobank	31,127 cases in men, 16,286 cases in women	Age, assay array, and 20 principal components	Individual level data in UK Biobank
Type 2 diabetes	DIAGRAM	41,846 cases in men, 30,053 cases in women	Sex, study-specific covariates, population structure and relatedness	[27]
Prostate cancer	PRACTICAL	79,148 cases, 61,106 controls	Principal components and study relevant covariates	[28]
Breast cancer	Combined Oncoarray; iCOGS; GWAS meta-analysis	122,977 cases, 105,974 controls	Principal components, and country	[29]
ER+ breast cancer		69,501 cases, 105,974 controls		[29]
ER- breast cancer		21,468 cases, 105,974 controls		[29]
Endometrial cancer	[30]	12,906 cases, 108,979 controls	Principal components	[30]
Ovarian cancer	OCAC	25,509 cases, 40,941 controls	Principal components	[31]
Secondary outcomes				
Overall				
LDL-c	GLGC (without UK Biobank)	Up to 842,544	Age, age^2^, principal components of ancestry and any necessary study-specific covariates	https://csg.sph.umich.edu/willer/public/glgc-lipids2021/ (accessed on 5 March 2024) [32]
Triglycerides	GLGC (without UK Biobank)	Up to 863,369	Age, age^2^, principal components of ancestry and any necessary study-specific covariates	https://csg.sph.umich.edu/willer/public/glgc-lipids2021/ (accessed on 5 March 2024) [32]
Apolipoprotein B	UK Biobank	440,546	Age, sex, age × sex, age^2^, age^2^ × sex and the first 10 principal components	[33]
Fasting glucose	MAGIC	140,595	Age, sex, BMI, and study-specific covariates	https://magicinvestigators.org/downloads/ (accessed on 4 May 2024)
HbA_1c_	MAGIC	145,579	Age, sex and study-specific covariates	[34]
BMI	UK Biobank	461,460	Age, sex, age × sex, age^2^, age^2^ × sex and the first 10 principal components	https://gwas.mrcieu.ac.uk/ (accessed on 4 May 2024)
SBP and DBP	GWAS meta-analysis of ICBP and UK Biobank	757,601	Age, age^2^, BMI, assay array	[35]
Sex-specific				
LDL-c and triglycerides	GLGC	Up to 686,787 in men, 570,542 in women	Age, age^2^, principal components of ancestry and any necessary study-specific covariates	https://csg.sph.umich.edu/willer/public/glgc-lipids2021/ (accessed on 5 March 2024) ([32])
Apolipoprotein B, glucose, HbA_1c_, BMI, SBP and DBP	UK Biobank (Neal Lab round 2)	167,020 men, 194,174 women	Age, age^2^, and the first 10 principal components	http://www.nealelab.is/uk-biobank (accessed on 4 May 2024)

## Data Availability

Data described in the manuscript will be available upon request and approval by the UK Biobank “https://www.ukbiobank.ac.uk/enable-your-research/apply-for-access (accessed on 21 February 2024)”. Other data are publicly available, with details of sources listed in Table 1.

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
