# Peer review of "Using Genetics to Assess the Role of Acetate in Ischemic Heart Disease, Diabetes, and Sex-Hormone-Related Cancers: A Mendelian Randomization Study"

_nutrients, 2024, doi:10.3390/nu16213674_

Round 1

Reviewer 1 Report

Comments and Suggestions for Authors

Starting from the role of acetate in the organism the benefits  and adverse effects are presented .

Acetate, a short chain fatty acid, has attracted increasing attention with dual activity, a potential benefit on cardiovascular health whilst concern on cancer, especially sex hormone-related cancers. Because these effects have been rarely examined in humans the main objective of the study was to determine the overall and sex-specific relationships between acetate and ischemic heart disease (IHD), diabetes, and sex hormone-related cancers.

The study design is very clear presented.

In the sex-specific analyses, the authors obtained  sex-specific genetic associations with acetate using individual data from the UK Biobank, and sex-specific genetic associations with the outcomes using UK Biobank or large consortia.

The results are very clear, presented in a logical manner.

The practical applicability is large.

I inserted few comments directly into the text.

Reference no 8 is very old

Comments on the Quality of English Language

Starting from the role of acetate in the organism the benefits  and adverse effects are presented .

Acetate, a short chain fatty acid, has attracted increasing attention with dual activity, a potential benefit on cardiovascular health whilst concern on cancer, especially sex hormone-related cancers. Because these effects have been rarely examined in humans the main objective of the study was to determine the overall and sex-specific relationships between acetate and ischemic heart disease (IHD), diabetes, and sex hormone-related cancers.

The study design is very clear presented.

In the sex-specific analyses, the authors obtained  sex-specific genetic associations with acetate using individual data from the UK Biobank, and sex-specific genetic associations with the outcomes using UK Biobank or large consortia.

The results are very clear, presented in a logical manner.

The practical applicability is large.

I inserted few comments directly into the text.

Reference no 8 is very old

Reviewer 2 Report

Comments and Suggestions for Authors

This paper evaluates the role of acetate in relation to IHD and cancer through Mendelian randomization, with the authors employing various statistical techniques to achieve appropriate corrections. It is a precisely conducted study; however, I believe there are several revisions needed to enhance the paper.

  1. Although the analysis and discussion of acetate's roles concerning IHD and cancer risk are included, they appear to be presented in a heterogeneous manner, making it difficult for readers to understand smoothly. Adding a graphical representation in Figure 1 that illustrates the relationship between the IV and the outcomes would aid in clearer communication of the main thesis.

  2. While this is mentioned in the limitations, the issue of overlap is significant when conducting two-sample MR. A clear description of what percentage of the patient cohort overlaps is necessary.

  3. Figures 3 and 4 lack labels for the axes, making it impossible to interpret those figures; therefore, corrections are needed.

Comments on the Quality of English Language

 Minor editing of English language required

Round 2

Reviewer 2 Report

Comments and Suggestions for Authors

 All comments have been answered appropriately.